# Selective Angiographic Roadmap Analysis (SARA) of Hepatocellular Carcinoma Feeding Arteries for Transarterial Chemoembolization

**DOI:** 10.3390/diagnostics15192533

**Published:** 2025-10-08

**Authors:** Sultan R. Alharbi

**Affiliations:** Department of Radiology and Medical Imaging, College of Medicine, King Saud University, Riyadh 11362, Saudi Arabia; drsultan000@gmail.com

**Keywords:** hepatocellular carcinoma, transarterial chemoembolization, hepatic arterial anatomy, accessory artery, cone-beam computed tomography, interventional radiology

## Abstract

Hepatocellular carcinoma (HCC) is a hypervascular malignancy commonly treated with transarterial chemoembolization (TACE), in which success relies on the accurate identification and embolization of tumor feeding arteries while sparing the nontumorous liver parenchyma. This review introduces the concept of selective angiographic roadmap analysis (SARA), a systematic and stepwise approach to evaluating hepatic arterial supply in HCC, with the aim of standardizing angiographic planning and improving TACE outcomes. SARA emphasizes recognition of typical and variant hepatic arterial anatomy, systematic identification of accessory and extrahepatic feeders, and integration with intraprocedural cone-beam computed tomography (CBCT) to enhance feeder detection and reduce nontarget embolization. Although primarily applied in TACE, the principles of SARA are equally relevant to transarterial radioembolization (TARE) where precise arterial mapping is critical. Embolization strategies are discussed across different levels of selectivity, from lobar to superselective techniques. The complementary role of advanced imaging modalities, such as CT angiography (CTA), MR angiography (MRA), and artificial intelligence-assisted vessel tracking, is also explored. Adopting the SARA framework in conjunction with these technologies may improve technical success and tumor control and preserve liver function in patients undergoing intra-arterial therapies.

## 1. Introduction

Hepatocellular carcinoma (HCC) is the most prevalent primary liver malignancy and ranks as the third leading cause of cancer-related deaths worldwide [1,2,3,4,5,6]. It often arises in the background of liver cirrhosis and hepatic fibrosis [7,8,9,10,11]. As hepatocarcinogenesis progresses, tumor neovascularization shifts the blood supply from the portal vein to the hepatic arteries, while the surrounding noncancerous liver parenchyma continues to be primarily supplied by the portal vein [12,13,14,15]. HCC is typically a hypervascular tumor, making it amenable to treatment through arterial embolization, which induces ischemia and necrosis of the tumor tissue [16,17,18].

Transarterial chemoembolization (TACE) delivers chemotherapeutic agents directly into a tumor’s feeding arteries, followed by arterial embolization to induce ischemia [19,20,21]. TACE is widely recommended as the first-line treatment for intermediate-stage HCC in several international clinical guidelines [22,23,24]. Its application has expanded across the spectrum of HCC management, from curative and bridging therapy to downstaging and palliative care [25,26]. Similarly, transarterial radioembolization (TARE) relies on precise arterial targeting to deliver radioactive microspheres and has shown increasing use in selected patients [27].

Despite its widespread use, TACE outcomes can be highly variable because of differences in patient selection, tumor burden, and technical execution [28,29,30]. Procedural success depends significantly on the complete embolization of all tumor feeding arteries while avoiding nontarget embolization. Thus, a standardized technical approach is essential for consistent and effective treatment [31,32]. Existing descriptions of hepatic arterial anatomy, such as the classical Michels and Hiatt classifications, provide a valuable foundation for understanding vascular variants [33]. However, these frameworks do not offer a procedural roadmap for intraprocedural decision-making during embolization. Operators continue to rely on personal experience and variable strategies for arterial mapping, which limits reproducibility and complicates communication.

This review presents a comprehensive analysis and classification system for angiographic roadmap planning during TACE, introducing Selective Angiographic Roadmap Analysis (SARA) as a systematic framework designed to address this gap. The goal of SARA is not to redefine anatomy but to standardize arterial mapping in HCC cases, enhance feeder identification, and minimize nontarget embolization, thereby improving intra-arterial therapy outcomes.

## 2. Arterial Anatomy Mapping and Classification

All patients undergoing TACE should undergo preprocedural contrast-enhanced computed tomography (CT) or magnetic resonance imaging (MRI) to confirm an HCC diagnosis. Given the common anatomical variations in the celiac axis, superior mesenteric artery (SMA), and hepatic arteries, meticulous vascular mapping before TACE is crucial [34,35,36,37,38]. During TACE, angiographies of the celiac trunk and the SMA provide vital information on hepatic arterial flow, particularly in cases of vascular stenosis or occlusion [39]. Failure to recognize such variations or identify all tumor feeders may result in incomplete embolization and suboptimal tumor response [36,40].

Computer tomography angiography (CTA) provides excellent spatial resolution, rapid acquisition, and multiplanar/3D reconstruction, making it highly effective for identifying hepatic arterial variants and accessory feeders [41]. Magnetic resonance imaging angiography (MRA) offers high-quality vascular imaging without ionizing radiation and is particularly useful in patients with renal dysfunction or contraindications to iodinated contrast [42]. Both modalities complement digital subtraction angiography (DSA) by allowing noninvasive preprocedural mapping of variant anatomy and feeding arteries, thereby supporting selective angiographic roadmap analysis (SARA) planning [39].

Classical hepatic arterial anatomy has been extensively described by Michels and Hiatt, whose classifications detail common variants, such as a replaced right hepatic artery from the SMA or a replaced left hepatic artery from the left gastric artery [36,40,43]. These anatomical frameworks remain the basis of clinical practice. However, they do not provide a standardized intraprocedural decision-making roadmap for embolization. SARA builds on these established classifications by organizing arterial mapping into a functional framework that links anatomical variants to feeder identification, catheterization strategy, and embolization planning. The tumor’s arterial supply can be assessed through CTA/MRA and digital subtraction angiography (DSA), and it can be classified based on the following:The number of feeding segmental arteries (single vs. multiple).The origin of the feeders (unilobar vs. bilobar).Lobar hepatic artery anatomy (normal vs. variant).The patency of mesenteric arteries (celiac and/or SMA).The presence of accessory feeders from the following:Hepatic arteries;Mesenteric arteries;Extra-mesenteric arteries.

For educational purposes, we simply classified hepatocellular carcinoma feeding arteries into four categories, as listed in Table 1 and illustrated in Figure 1.

Certain variants carrying direct technical consequences may require an alternative catheterization strategy. For example, a replaced right hepatic artery arising from the SMA may require a hydrophilic wire or a more supportive guiding catheter to overcome acute angulation. Tortuous or diminutive accessory feeders may necessitate the use of low-profile thinner microcatheters and smaller hydrophilic microwires for superselective access. Awareness of such variants is critical not only to ensure complete tumor coverage but also to minimize the risk of nontarget embolization [39,43].

This conceptual framework does not aim to replace epidemiological data on variant prevalence but rather to standardize how feeders are described and approached during embolization. Its primary value is educational and procedural, providing a structured roadmap for feeder identification that can be validated prospectively in future studies correlating variant prevalence, technical difficulty, and clinical outcomes.

## 3. Embolization Point Classification

Successful TACE depends on the accurate identification and catheterization of tumor feeding arteries while sparing the remaining liver parenchyma [44]. Local tumor control and the preservation of liver function are key prognostic factors in HCC management [45]. Embolization strategies are typically classified into three categories: lobar, segmental, and subsegmental TACE (Table 2) [46].

The choice of embolization level depends on the number and accessibility of feeding arteries, which may be affected by anatomical complexity, tortuosity, or unfavorable angulation [47].

Generally, the choice of the embolization point is determined by the number and distribution of tumor feeding arteries, balancing the goals of complete tumor coverage with preservation of nontumorous parenchyma. As summarized in Table 3 and illustrated in Figure 2, superselective embolization is preferred when a single subsegmental or segmental feeder can be accessed, maximizing on-target ischemia while sparing uninvolved liver. In contrast, when multiple feeders arise from a single lobar artery, lobar embolization may be required to ensure complete treatment, albeit at the cost of a larger ischemic territory. Similarly, tumors supplied by feeders from two distinct lobar arteries necessitate two separate embolization points to achieve comprehensive coverage. The schematic use of target signs highlights this decision-making process within the SARA framework, emphasizing that embolization point selection is an anatomically driven strategy aimed at optimizing tumor control and minimizing nontarget embolization.

Superselective TACE is associated with higher tumor necrosis rates and improved survival compared with lobar approaches, which risk larger nontarget embolization and hepatic dysfunction [48,49]. Superselective TACE should be attempted to maximize tumor necrosis and minimize nontarget embolization to the remaining liver parenchyma [49].

## 4. Embolization of the Principal Feeding Artery

Effective embolization entails treating all identifiable tumor feeding arteries to near stasis [17]. Superselective embolization reduces nontarget effects; however, it may miss minor feeders and result in incomplete therapy [44]. Collateral arteries that are initially nondominant may become significant post-embolization because of hemodynamic redistribution [50]. Thus, performing comprehensive embolization of all primary and accessory feeders while maintaining perfusion to healthy liver tissue is essential [51].

Multiple studies have highlighted the critical role of achieving a “safety margin” during TACE to ensure complete tumor necrosis [44,52,53]. This principle, which is well established in curative treatments, such as surgical resection and thermal ablation, is increasingly recognized in TACE [54,55]. It involves embolizing not only the main tumor feeding arteries but also the peritumoral parenchymal arteries that may harbor microscopic disease, as shown in Figure 3 [56].

Histopathological analyses have shown the presence of microsatellite lesions in almost half of tumors measuring ≥5 cm and in one-third of tumors measuring <2.5 cm, with nearly all such lesions located within 0.5 cm of the primary tumor. The persistence of microsatellites or capsular invasion can lead to peritumoral recurrence after TACE [57,58]. The area of corona enhancement observed on imaging is believed to correspond to this peritumoral zone and may serve as a target for extending the embolization margin [56]. Effectively identifying and embolizing these adjacent feeders is essential, as the ischemic tumor bed may recruit them post-treatment, potentially resulting in tumor progression or recurrence [44].

## 5. Accessory Arterial Supply

Accessory arterial supply refers to any feeding artery other than the principal feeder that contributes to tumor perfusion, as illustrated in Figure 4. Unrecognized accessory feeding arteries are a common cause of incomplete embolization and a suboptimal tumor response [59]. Hence, a careful analysis of preprocedural imaging and intraprocedural angiography is crucial.

Factors associated with an increased probability of accessory arterial supply include the following [39,40,60]:(A)Tumor size, with >5 cm increasing the risk of ectopic arterial supply.(B)Tumor location:Exophytic: Larger tumor surface area exposed to adjacent organ arterial supply;Subcapsular: Near to the adjacent organ arterial supply;Perihilar: Near to multiple hepatic and mesenteric artery branches;Segments 4 and 1: Watershed areas between the right and left hepatic arteries.(C)Repeated TACE, which may occlude primary feeders and promote collateral formation.(D)Partial visualization of tumors on hepatic angiography, which may indicate accessory extrahepatic supply to non-visualized parts of the tumor.

Accessory feeders can be classified as follows:(A)Accessory hepatic arteries: Common hepatic artery variations include an accessory left hepatic artery from the left gastric artery, which may supply left liver lobe HCC, and an accessory right hepatic artery from the SMA, which may supply right liver lobe HCC. Segments 1 and 4 frequently receive dual supply from both lobar hepatic arteries. An accessory middle hepatic artery may arise from the left or right hepatic artery or from the hepatic proper artery as a trifurcation branching. Accessory middle hepatic arteries usually supply segment 4 but may contribute to the supply of right or left liver lobe HCC [36,40,61].(B)Accessory mesenteric arteries: HCC may be fed by mesenteric branches, such as the left gastric, gastroduodenal, superior mesenteric, and pancreaticoduodenal arteries. Left lobe tumors may recruit the left gastric or pancreaticoduodenal arteries, whereas right lobe tumors may be supplied by gastroduodenal and omental branches [62,63].(C)Accessory extra-mesenteric arteries: Tumors that are located anteriorly at liver dome segments 2, 4A, and 8 may receive accessory supply from the internal mammary artery, a branch of the right subclavian artery. However, HCC located posteriorly at liver dome segments 8, 7, and 2 and the caudate may receive accessory supply from the inferior phrenic artery, a branch of the celiac trunk or directly originating from the abdominal aorta [64]. A subcapsular tumor near the ribs may receive accessory supply from intercostal arteries, as shown in Figure 5 [65].

To achieve effective transarterial chemoembolization (TACE), it is essential to identify and embolize not only the hepatic arterial branches but also any accessory or extrahepatic collateral arteries supplying the HCC [49,55,66]. Failure to treat these additional feeding arteries can result in incomplete embolization, a suboptimal tumor response, and poorer prognosis [54]. Therefore, selective angiography should be performed to evaluate suspected extrahepatic collaterals, particularly when pre-treatment imaging reveals variant anatomy or hypertrophied collateral arteries, enabling comprehensive tumor coverage and improving the likelihood of a complete local tumor response [39].

## 6. Additive Role of Cone-Beam CT

C-arm cone-beam CT (CBCT), which utilizes flat-panel detector technology, has become an invaluable tool for intraprocedural imaging in interventional radiology. It offers a high-resolution, three-dimensional visualization of tumors and their vascular supply. In the setting of TACE, CBCT enhances the detection of small HCC lesions, tumor feeding arteries, and extrahepatic collateral arteries, and it predicts the treatment response [59,67]. Compared to conventional digital subtraction angiography (DSA), CBCT demonstrates superior sensitivity and accuracy in identifying tumor feeders, as shown in Figure 6 [68,69].

This advanced imaging modality can influence the treatment strategy in nearly half of TACE cases, thereby optimizing the procedure and contributing to an improved tumor response and patient survival outcomes [70].

Within selective angiographic roadmap analysis (SARA), CBCT provides intraprocedural three-dimensional arterial mapping that supplements DSA, allowing for precise identification of small and accessory feeders that might otherwise be overlooked. Multiphase CBCT can be employed both before embolization for detailed arterial mapping and after embolization to assess the adequacy of treatment and detect any residual tumor perfusion, as illustrated in Figure 7 [71].

By integrating CBCT into angiographic planning, SARA enables more comprehensive feeder assessment, improves embolization selectivity, and reduces the risk of undertreatment, as described in Table 4.

The integration of CBCT into TACE protocols has been shown to significantly reduce local tumor recurrence, prolong overall survival, and improve local progression-free survival. Due to these clear clinical benefits, the use of CBCT during TACE is strongly endorsed by international guidelines as a standard component of a high-quality TACE procedure [20,39,72,73,74].

## 7. Additive Role of Artificial Intelligence

C-arm cone-beam CT (CBCT) enables interventional radiologists to acquire real-time cross-sectional and three-dimensional (3D) images during the TACE procedure. These images serve as navigational tools, providing 3D road mapping that facilitates precise catheter guidance and superselective embolization. This advanced vessel navigation significantly improves the detection of tumor nodules and their feeding arteries. In particular, volume-rendered 3D CBCT images can clearly delineate tumor feeding arteries, even when overlaid by naïve hepatic arteries [44,59]. More recently, artificial intelligence (AI)-based automated feeder detection (AFD) software—such as EmboGuide^®^ and FlightPlan^®^—has been integrated with CBCT imaging in the TACE workflow. These tools enhance the identification of subsegmental tumor feeders, achieving detection rates as high as 81–93%, which surpasses the performance of CBCT interpretation alone, as shown in Figure 8 [31,44,53,72].

As automated feeder detection (AFD) software is built upon the CBCT platform, it is expected to offer at least the same capability as conventional CBCT in preserving healthy liver parenchyma. Additionally, it enhances the detection of small intrahepatic accessory feeding vessels, helping to prevent tumor recurrence due to previously unidentified collaterals [44,75]. Furthermore, second-generation AI tools, such as virtual parenchymal perfusion software based on CBCT, can estimate the expected therapeutic coverage and predict the embolization territory. This allows for more precise targeting, minimizes nontarget embolization, and ultimately improves procedural outcomes while preserving liver function, as illustrated in Figure 9 [76,77].

## 8. Summary of Stepwise Workflow of SARA

The SARA framework begins before the procedure and extends into intraprocedural decision-making. Artificial intelligence (AI) serves as an adjunctive tool throughout this process, enhancing—but not replacing—the operator’s interpretation.

Step 1: Preprocedural imaging review

Contrast-enhanced CT or MRI defines tumor size, number, and location. CTA or MRA further delineates hepatic arterial anatomy and variants, forming the basis of an anatomical roadmap.

Step 2: Angiographic roadmap construction

Initial DSA of the celiac, superior mesenteric, and hepatic arteries confirms expected feeders and identifies accessory or variant supply. This roadmap is compared with the preprocedural anatomical map.

Step 3: CBCT confirmation and refinement

CBCT acquisitions at lobar or segmental levels provide three-dimensional visualization of tumor enhancement and arterial supply. Technical parameters are adapted to catheter position, ensuring optimal lesion delineation while minimizing artifact.

Step 4: AI-assisted analysis of CBCT

AI software processes CBCT datasets to

Automatically extract and reconstruct the hepatic arterial tree;Highlight potential tumor feeders, even when overlying vessels obscure direct visualization;Provide a “virtual roadmap” for catheter trajectory planning.

Step 5: Virtual embolization planning

AI-driven perfusion simulation tools predict the vascular territory affected if embolization is performed at a selected point. This allows operators to visualize whether the target tumor will be fully covered, while simultaneously estimating the extent of nontumorous parenchyma at risk.

Step 6: Informed embolization execution

Armed with both conventional roadmap analysis and AI-generated predictions, the operator selects the most effective embolization point. The embolization is then carried out with greater confidence that feeders are complete and parenchymal preservation is maximized.

Step 7: Post-embolization validation

Post-embolization DSA and CBCT can be re-processed with AI tools to confirm devascularization, assess residual enhancement, and ensure consistency between planned and actual treatment coverage.

## 9. Conclusions

Transarterial chemoembolization remains a cornerstone in the management of hepatocellular carcinoma. Its success relies heavily on accurate arterial mapping and the complete embolization of tumor feeding arteries while minimizing injury to the surrounding liver parenchyma. The proposed selective angiographic roadmap analysis (SARA) framework provides a structured approach to classifying hepatic arterial anatomy, embolization points, and accessory arterial supply.

Advancements in imaging technologies, such as CBCT- and AI-based vessel tracking and virtual parenchymal perfusion, have further enhanced the precision of TACE, allowing for more personalized and effective treatment strategies. A standardized and meticulous angiographic approach can improve the tumor response, reduce recurrence, and ultimately enhance overall patient outcomes. Further prospective studies are needed to validate the clinical utility of the SARA framework in improving TACE outcomes across various HCC presentations.

## Figures and Tables

**Figure 1 diagnostics-15-02533-f001:**
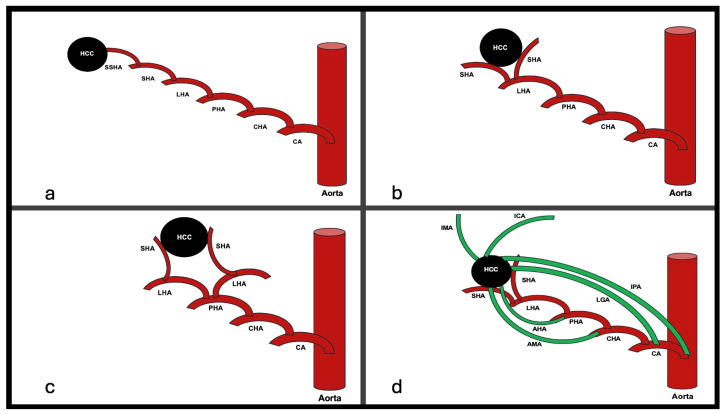
Classification of HCC feeding arteries. (**a**) Type 1. Single feeding artery branch of single lobar hepatic artery. (**b**) Type 2. Multiple feeding arteries from single lobar hepatic artery. (**c**) Type 3. Multiple feeding arteries from multiple lobar hepatic arteries. (**d**) Type 4. Multiple feeding arteries from lobar hepatic arteries plus accessory arteries. Green colored arteries indicate accessory feeders. CA: celiac artery; CHA: common hepatic artery; PHA: proper hepatic artery; LHA: lobar hepatic artery; SHA: segmental hepatic artery; SSHA: subsegmental hepatic artery; IPA: inferior phrenic artery; LGA: left gastric artery; AMA: accessory mesenteric artery; AHA: accessory hepatic artery; IMA: internal mammary artery; ICA: intercoastal artery.

**Figure 2 diagnostics-15-02533-f002:**
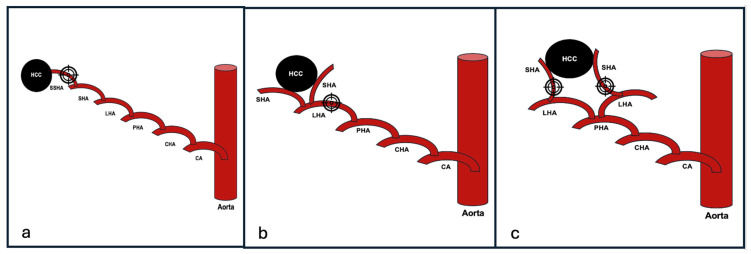
(**a**) Subsegmental embolization point; (**b**) lobar embolization point; (**c**) two segmental embolization points. Target signs indicate the recommended embolization site in each scenario.

**Figure 3 diagnostics-15-02533-f003:**
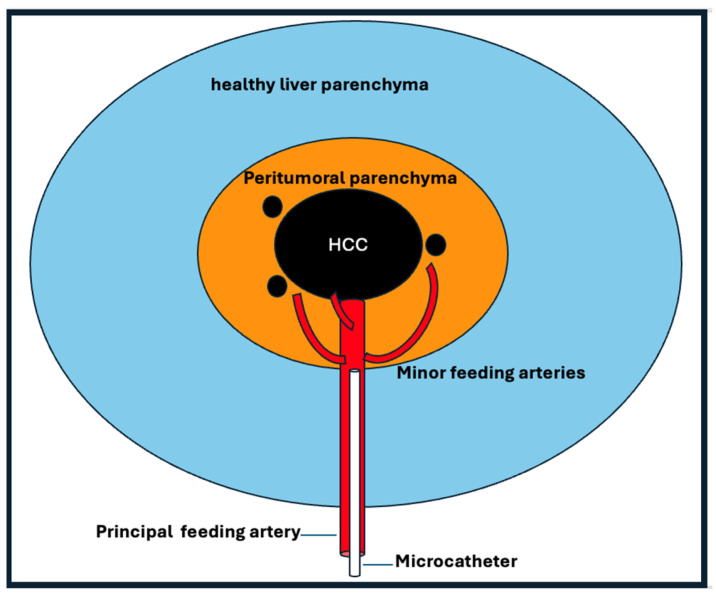
Embolization of the principal feeding artery should include the minor feeding arteries in the peritumoral area. The peritumoral area may contain microsatellite lesions.

**Figure 4 diagnostics-15-02533-f004:**
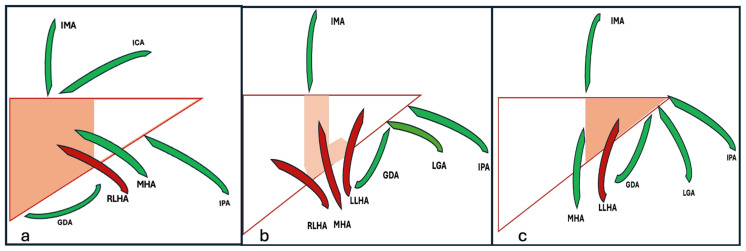
Common accessory arterial supply. (**a**) Right hepatic lobe, (**b**) segments 4 and 1, and (**c**) left liver lobe. RLHA: right lobar hepatic artery; LLHA: left lobar hepatic artery; MHA: middle hepatic artery; GDA: gastroduodenal artery; LGA: left gastric artery; IPA: inferior phrenic artery; IMA: internal mammary artery; ICA: intercostal artery.

**Figure 5 diagnostics-15-02533-f005:**
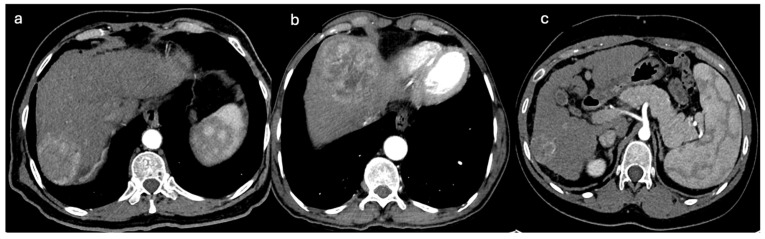
Axial CT. (**a**) CT image showing HCC located posteriorly at liver dome with an accessory feeding supply from inferior phrenic artery. (**b**) CT image showing HCC located anteriorly at liver dome with an accessory feeding supply from internal mammary artery. (**c**) CT image showing subcapsular HCC near the rib with an accessory feeding supply from intercoastal artery.

**Figure 6 diagnostics-15-02533-f006:**
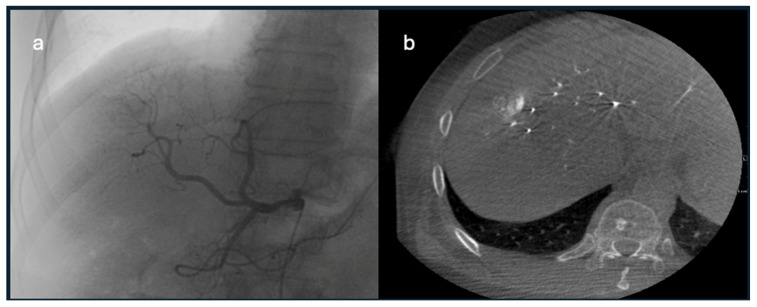
CBCT during TACE. (**a**) DSA image of HCC feeding artery. (**b**) CBCT image showing enhancing HCC.

**Figure 7 diagnostics-15-02533-f007:**
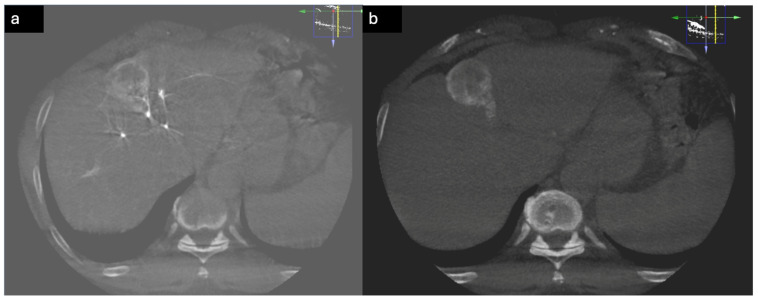
CBCT during TACE. (**a**) CBCT image of HCC before embolization showing enhancing lesion with feeding artery. (**b**) CBCT after embolization image (without contrast) showing complete coverage of HCC.

**Figure 8 diagnostics-15-02533-f008:**
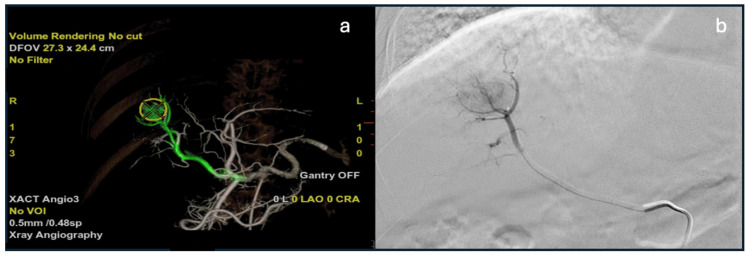
Automated feeder detection software. (**a**) Volume rendering image of software feeder detection showing a segmental arterial HCC feeder from celiac angiography. (**b**) DSA image of selected segmental feeder branch showing a hypervascular HCC.

**Figure 9 diagnostics-15-02533-f009:**
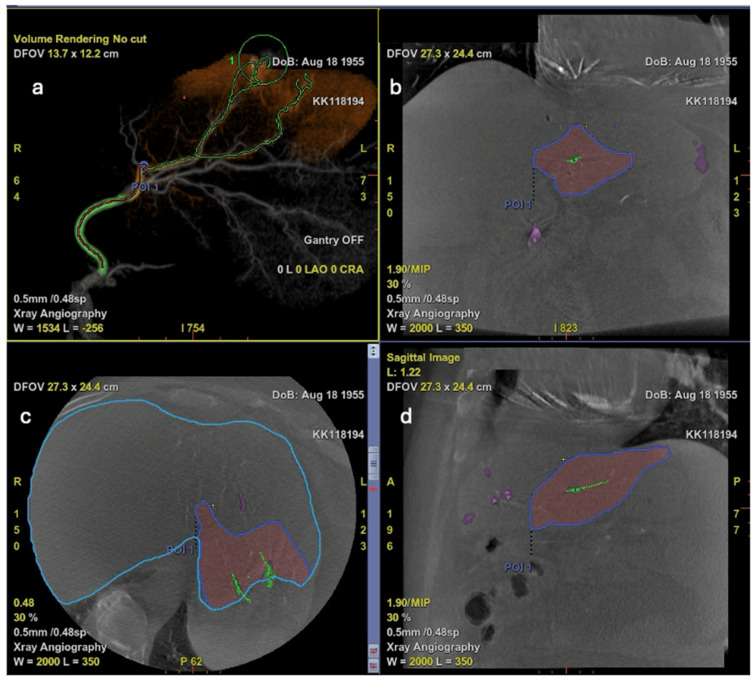
Virtual parenchymal perfusion software images. (**a**) Volume rendering image of automated detection of HCC feeders. (**b**) Coronal image of affected liver parenchyma during TACE. (**c**) Axial image of affected liver parenchyma during TACE. (**d**) Sagittal image of affected liver parenchyma during TACE. Affected liver parenchyma is colored and labeled by point of interest (POI).

**Table 1 diagnostics-15-02533-t001:** Classification of hepatocellular carcinoma feeding arteries.

Type	Description
1	Single feeding artery branch of a single lobar hepatic artery
2	Multiple feeding arteries from a single lobar hepatic artery
3	Multiple feeding arteries from multiple lobar hepatic arteries
4	Multiple feeding arteries from lobar hepatic arteries plus accessory arteries

**Table 2 diagnostics-15-02533-t002:** Embolization point types, locations of catheter tip, and affected liver parenchyma.

Type	Catheter Tip Location	Hepatic Parenchyma Affected
Lobar TACE (non-selective)	Lobar hepatic artery	Whole liver lobe
Selective TACE	Segmental hepatic artery	Liver segment
Superselective TACE	Subsegmental hepatic artery	Part of liver segment

**Table 3 diagnostics-15-02533-t003:** Embolization points for each feeding artery classification.

Feeding HCC Artery	Embolization Point
Single feeding artery from single lobar artery	Superselective or selective (one point)
Multiple feeding arteries from one lobar artery	Lobar embolization point
Multiple feeding arteries from two lobar arteries	Superselective or selective (two points)

**Table 4 diagnostics-15-02533-t004:** CBCT utilization in different scenarios during TACE.

Scenario	Purpose
1. No tumor blush or feeder artery seen in DSA	To detect subtle feeders and confirm subtle less vascular tumors
2. Overlying multiple arteries	To ensure complete tumor coverage/guide selectivity
3. Suspected accessory feeder arteries	To identify accessory or extrahepatic tumor supply
4. Post-embolization	To verify devascularization and technical success

## Data Availability

Not applicable.

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
