# Peer review of "Selective Angiographic Roadmap Analysis (SARA) of Hepatocellular Carcinoma Feeding Arteries for Transarterial Chemoembolization"

_diagnostics, 2025, doi:10.3390/diagnostics15192533_

Round 1
Reviewer 1 Report
Comments and Suggestions for Authors
This paper proposes a classification framework for types of hepatic arteries and their area of supply of an HCC, to standardize the nomenclature of the TACE procedure.
This is how the procedure is generally performed, and therefore, the classification does not add value to the field. The classification should be based on the known classification of hepatic arterial anatomy, which is well described.
There is no data to attribute the relative importance of each anatomical variation and its implications for the procedure, such as what types of guidewires may be required for a certain classification. Although the proposed classification may be very useful, without data about the prevalence of each type found in patients, it is of little use and mostly theoretical.
Author Response
Comments 1: [This is how the procedure is generally performed, and therefore, the classification does not add value to the field.]
Response 1: [ I appreciate this perspective. I have clarified in the revised manuscript that SARA is not intended to replace existing hepatic arterial classifications, but to provide a structured, stepwise angiographic framework that standardizes the procedural thought process during TACE. Its novelty lies in synthesizing established anatomical descriptions into a practical roadmap for planning and execution. Changes can be found-page 2, paragraph 3,4 and lines 52-64 ]
Comments 2 :[The classification should be based on the known classification of hepatic arterial anatomy, which is well described.]
Response 2 : [ agree and I have expanded the "Arterial anatomy mapping and classification ” section to explicitly reference established classifications (Michels, Hiatt), showing how SARA integrates these into a functional, procedure-oriented roadmap.Changes can be found-page 3, paragraph 2 and lines 79-89 ]
Comments 3: [There is no data to attribute the relative importance of each anatomical variation and its implications for the procedure, such as what types of guidewires may be required for a certain classification.]
Response 3 : [I have added examples demonstrating how specific arterial variants (e.g., replaced right hepatic artery from SMA, tortuous accessory feeders) may necessitate distinct catheter/wire strategies. I also highlight the importance of recognizing variants for minimizing nontarget embolization. While data on prevalence and technical difficulty are currently lacking, I propose that future studies validate SARA by correlating variant types with catheterization strategies, procedural outcomes, and recurrence rates. Changes can be found- page 4, paragraph 1 and lines 110-115]
Comments 4: [Although the proposed classification may be very useful, without data about the prevalence of each type found in patients, it is of little use and mostly theoretical.]
Response 4: [I acknowledge this limitation and have added a paragraph in "Arterial anatomy mapping and classification ” section to clarify that SARA is an educational and conceptual framework designed to standardize angiographic planning. I believe that establishing the framework is a necessary first step before prospective validation studies can assess prevalence and procedural impact. Change can be found-page 4, paragraph 2 and lines 116-120]
Reviewer 2 Report
Comments and Suggestions for Authors
Dear Authors,
the Manuscript is very interesting and useful for daily clinical practice.
Author Response
comments: 1 [the Manuscript is very interesting and useful for daily clinical practice.]
Response: 1 [I sincerely thank the reviewer for their positive feedback and for recognizing the clinical usefulness of my work.]
Reviewer 3 Report
Comments and Suggestions for Authors
This study is important, timely and well done. You conclude that more prospective studies are needed to validate your results in patients. Are you collaborating with other radiologic centers around the world who are undoubtedly interested?
Author Response
Comments 1: [This study is important, timely and well done. You conclude that more prospective studies are needed to validate your results in patients. Are you collaborating with other radiologic centers around the world who are undoubtedly interested? ]
Response 1: [ I sincerely thank the reviewer for the encouraging feedback. I fully agree that prospective validation across multiple centers is essential to confirm our findings. At present, this study is single-center; however, I am actively exploring the possibility of establishing collaborations with other interventional radiology centers. I strongly believe that multicenter cooperation will enhance the generalizability of my results.]
Reviewer 4 Report
Comments and Suggestions for Authors
There are no specific remarks. the topic is important and novel, it is clearly demonstrated. It would be of great interest for readers
Author Response
Comments 1: [There are no specific remarks. the topic is important and novel, it is clearly demonstrated. It would be of great interest for readers]
Response 1: [I sincerely thank the reviewer for their positive feedback and encouraging remarks. I am pleased that the novelty and clarity of my work were recognized and believe readers will also find it valuable.]
Reviewer 5 Report
Comments and Suggestions for Authors
This review introduces the concept of selective angiographic roadmap analysis (SARA), which is a systematic method for assessing hepatic arterial supply in hepatocellular carcinoma (HCC). The goal of SARA is to standardize angiographic planning and enhance the outcomes of transarterial chemoembolization (TACE). The review discusses the classification of hepatic arterial anatomy, including both typical and variant patterns, and examines the number and origin of feeding arteries. However, there are several concerns that need to be addressed before publication.
- Abstract: The important features of SARA should be added.
- Introduction: a) The clinical problem that leads to the use of SARA should be explained. b) Since SARA is used in TARE procedures, it should be mentioned.
- Arterial Anatomy Mapping and Classification: The technical features of CTA and MRA should be explained in a separate paragraph.
- Additive role of cone-beam CT and artificial intelligence: a) Since the artificial intelligence topic already exists, it should be changed to the additive role of cone-beam CT. b) Technical details of cone-beam CT in terms of SARA should be added.
Author Response
Comments 1: [Abstract: The important features of SARA should be added.]
Response 1: [Thank you for this suggestion. I have revised the abstract to explicitly highlight the stepwise features of SARA, including evaluation of variant anatomy, systematic identification of accessory feeders, and integration with CBCT. Change can be found-page 1, paragraph 1 and lines 15-19 ]
Comments 2: [The clinical problem that leads to the use of SARA should be explained]
Response 2: [ agree and I have expanded the introduction to discuss the angiographic challenges (complex hepatic anatomy, accessory feeders, risk of nontarget embolization) that motivated the development of SARA. Changes can be found-page 2 paragraph 2,3 and lines 52-63]
Comments 3: [Since SARA is used in TARE procedures, it should be mentioned.]
Response 3: [ Agree. I have added a note on the applicability of SARA to TARE procedures, where accurate mapping is equally critical. Changes can be found-page 2, paragraph 2 and lines 44-46]
Comments 4: [ Arterial Anatomy Mapping and Classification: The technical features of CTA and MRA should be explained in a separate paragraph.]
Response 4: [Agree. I have added a dedicated paragraph describing the technical role of CTA and MRA in mapping hepatic arterial anatomy, highlighting their complementary role to angiography. Changes can be found-page 2,3, paragraph 2 and lines 75- 82]
Comments 5: [Additive role of cone-beam CT and artificial intelligence: Since the artificial intelligence topic already exists, it should be changed to the additive role of cone-beam CT.]
Response 5: [ Agree. I have restructured this section under the heading “Additive Role of Cone-Beam CT ”. Changes can be found-page 8, line 237]
Comments 6: [Technical details of cone-beam CT in terms of SARA should be added.]
Response 6: [Agree. Technical details of CBCT in feeder detection, multiphase imaging, and integration into SARA angiographic planning have been added. Change can be found-page 9, paragraph 2,3 lines 252-254, 262-266]
Round 2
Reviewer 1 Report
Comments and Suggestions for Authors
The manuscript has been appropriately altered to include existing anatomical classifications and justification for lack of patient data in this paper.
Reviewer 5 Report
Comments and Suggestions for Authors
The essential changes have been successfully implemented, and I have no additional comments to make.